# *Artemisia vulgaris* Extract as a Novel Therapeutic Approach for Reversing Diabetic Cardiomyopathy in a Rat Model

**DOI:** 10.3390/ph17081046

**Published:** 2024-08-08

**Authors:** Ghulam Hussain, Abdul Malik, Suhail Akhtar, Haseeb Anwar

**Affiliations:** 1Department of Physiology, Faculty of Life Sciences, Government College University, Faisalabad 38000, Pakistan; lizamalik981@gmail.com (L.); ghulamhussain@gcuf.edu.pk (G.H.); 2Department of Pharmaceutics, College of Pharmacy, King Saud University, Riyadh 11564, Saudi Arabia; amoinuddin@ksu.edu.sa; 3Department of Biochemistry, A.T. Still University of Health Sciences, Kirksville, MO 63501, USA; suhailakhtar@atsu.edu

**Keywords:** *Artemisia vulgaris*, diabetes, diabetic cardiomyopathy, ECG

## Abstract

Diabetic cardiomyopathy, a severe diabetic complication, impairs heart function, leading to heart failure. Treatment that effectively addresses this condition without causing side effects is urgently needed. Current anti-hyperglycemic therapies are expensive, has side effects and do not effectively prevent cardiac remodeling. Therefore, it is important to explore natural products that may have the potential to reverse cardiac remodeling. That is why the aim of the current study was to determine the left ventricular remodeling potential of the methanolic extract of *Artemisia vulgaris* in a diabetic cardiomyopathy rat model. Following the initial comprehensive phytochemical evaluation of plant phenolic and flavonoid content, which showed strong anti-hyperglycemic and antioxidant activities, an extract of *Artemisia vulgaris* was administered in an in vivo experiment. Diabetic cardiomyopathy was induced in Wistar albino rats according to previously described protocols in the literature, and the effect of treatment was checked by serum and histopathological analysis after 45 days. *Artemisia vulgaris* treatment significantly (*p* ≤ 0.05) reduced fasting blood glucose (108.5 ± 1.75 mg/dL), glycated hemoglobin (4.03 ± 0.12 %), serum glucose (116.66 ± 3.28 mg/dL), insulin (15.66 ± 0.66 ng/mL), total oxidant status (54.66 ± 3.22 µmol H_2_O_2_Equiv.L^−1^), Malondialdehyde (0.20 ± 0.01 mmol/L), total cholesterol (91.16 ± 3.35 mg/dL), triglycerides (130.66 ± 3.15 mg/dL), low-density lipids (36.57 ± 1.02 mg/dL), sodium (140 ± 3.21 mmol/L), calcium (10.44 ± 0.24 mmol/L), creatine kinase MB (1227.5 ± 17.89 IU/L), lactate dehydrogenase (1300 ± 34.64 IU/L), C-reactive protein (30 ± 0.57 pg/mL), tumor necrosis factor-α (58.66 ± 1.76 pg/mL), atrial natriuretic peptide (2.53 ± 0.04 pg/mL), B-type natriuretic peptide (10.66 ± 0.44 pg/mL), aspartate aminotransferase (86.5 ± 4.99 U/L), Alanine Transaminase (55.33 ± 2.90 U/L), urea (25.33 ± 1.15 mg/dL) and creatinine (0.64 ± 0.02 mg/dL) but significantly increased (*p* ≤ 0.05) total antioxidant capacity (1.73 ± 0.07 mmol Trolox Equil./L), high-density lipids (40 ± 1.59 mg/dL) and potassium (3.82 ± 0.04 mmol/L) levels. ECG and histopathology confirmed the significant improvement in remodeling and the reversal of structural changes in the heart and pancreas. In conclusion, *Artemisia vulgaris* possesses significant left ventricular remodeling potential in course of diabetes-induced cardiomyopathy.

## 1. Introduction

According to the International Diabetes Federation, in 2022, diabetes affected approximately 26.7% of adults in Pakistan, totaling approximately 33 million cases [1]. Diabetes is a leading cause of cardiac mortality, blindness and renal failure with over 463 million people living with this condition worldwide. According to research, 600 million people will suffer from diabetes by the end of 2035, and this number could reach 700 million by 2045 [2]. Diabetes complicates body function and predominantly enhances cardiovascular complications, frequently leading to heart failure. Compared to nondiabetic people, diabetic people experience a two to five fold increase in the development of cardiovascular complications [3]. This condition is characterized by early-onset diastolic and late-onset systolic dysfunction and is known as diabetic cardiomyopathy (DCM), which is characterized by functional and structural abnormalities in the heart without any other traditional risk factors, including severe valvular disease, ischemic heart disease or hypertension. Approximately 40–75% of diabetic patients [4] and rodent diabetic models have diastolic dysfunction [5] without the presence of atherosclerosis or vascular/valvular dysfunction signifies a specific response of diabetes toward the heart [6].

The causes of DCM include complex factors such as hyperglycemia, insulin resistance, hyperinsulinemia, lipotoxicity and metabolic changes via several mechanisms. The discernable effects of DCM include fibrosis, left ventricular hypertrophy and a decrease in systolic and diastolic dysfunction which leads to damage of cardio myocytes and release of cardiac enzymes, natriuretic peptides, mitochondrial and cellular calcium mishandling, and increased oxidative stress and inflammation. So, these are the parameters which serve as a marker to assess left ventricular dysfunction [7,8]. Despite wide-ranging research, there is no specific medication for DCM, and the available anti-hyperglycemic therapies do not completely reverse cardiac remodeling and have side effects; thus, the possibility of natural products with the potential to reverse cardiac remodeling as healthier treatments should be explored. Phytotherapy might offer a promising alternative to pharmaceuticals in managing diabetic cardiomyopathy, providing natural compounds that mitigate oxidative damage and inflammation while supporting cardiovascular health. The isolation of iminosugars from natural sources for drug development in diabetes management has garnered significant interest in recent years. These compounds, known for their ability to inhibit alpha-glucosidase enzymes, illustrate the compelling transition from natural origins to synthetic drug development pathways [9]. Therefore, our study assessed the phytotherapy of *Artemisia vulgaris* roots for early DCM left ventricular remodeling. *Artemisia vulgaris* is native to Asia, Europe, Alaska and North Africa [10]. It holds culinary significance and is frequently utilized as a vegetable or included in soups across Asia [11]. Roots are also used for medicinal purposes [10] and were chosen as raw material because they have high phenolic and flavonoid content, and greater antioxidant and antidiabetic potential as compared to aerial parts [12]; in addition, they have hypolipidemic effects [13].

## 2. Results

### 2.1. In Vitro Phytochemical Analysis of the Prepared Plant Extracts

During the initial phytochemical screening, the methanolic extract of *Artemisia vulgaris* contained different phytochemicals, primarily phenolic acids, alkaloids and flavonoids, as shown in Table 1. However, the total phenolic content (TPC) and total flavonoid content (TFC) were 24.03 ± 2.64 (mg/g GAE) and 16.16 ± 0.72 (mg/g QE), respectively. An HPLC analysis confirmed the presence of six phenolic compounds with p-coumaric acid found in abundance, as shown in Figure 1 and Table 2. GC–MS analysis revealed the presence of many antioxidant, anti-inflammatory and antidiabetic compounds, as shown in Figure 2 and Table 3. These compounds are mainly responsible for the cure of diabetic cardiomyopathy because DCM is aggravated by oxidative stress and inflammation; therefore, these antioxidant and anti-inflammatory compounds attenuates this condition. The antioxidant potential was further confirmed by determining the DPPH-radical-scavenging activity, which was 78.33 ± 1.65% at 5 mg/mL (Figure 3.) The antidiabetic activity was measured by an enzyme inhibition assay, and the percentages of inhibition of α-amylase and α-glucosidase were 75.96 ± 1.86% and 81.13 ± 2.14%, respectively, at 5 mg/mL (Figure 4). 

### 2.2. Glycemic Index

Fasting blood glucose, serum glucose, serum insulin and glycated hemoglobin (HbA1c) levels exhibited persistent increases in the PC group throughout the trial. As anticipated, metformin, a standard treatment (STD), effectively attenuated these levels. Similarly, *Artemisia vulgaris* treatment significantly reduced fasting blood glucose, serum glucose, serum insulin and HbA1c levels. The negative control (NC) displayed no significant variations until the conclusion, as illustrated in Figure 5. These results indicate a therapeutic potential of *Artemisia vulgaris* in lowering overall glycemic index.

### 2.3. Oxidative Stress Parameters

The outcomes pertaining to oxidative stress are shown in Figure 6. A one-way analysis of variance revealed noteworthy differences in the mean total oxidant status (TOS), total antioxidant capacity (TAC) and malondialdehyde (MDA) content among the diverse groups. Streptozotocin administration elevated oxidative stress, as evidenced by the TAC, TOS and MDA levels in the PC group. The standard (STD) and *Artemisia vulgaris*-supplemented groups exhibited markedly increased TAC levels, leading to significant reductions in TOS and MDA levels. Overall *Artemisia vulgaris* significantly reduced oxidative stress.

### 2.4. Serum Liver Enzymes 

The results regarding serum liver enzymes are shown in Figure 7, in which a significant increase in aspartate aminotransferase (AST) and alanine aminotransferase (ALT) levels in the PC group was observed. Both the standard treatment and *Artemisia vulgaris* treatment significantly reduced the AST and ALT levels, showing the reversal of liver function.

### 2.5. Renal Function Markers

After the administration of streptozotocin (STZ), elevated serum urea and creatinine levels were evident, particularly in the PC group due to damage to the heart which subsequently effects kidney and liver function. Treatment with the standard drug metformin significantly reduced urea and creatinine levels, while urea levels were near normal in the AVM group as detailed in Figure 8. 

### 2.6. Lipid Profile

A notable disparity in the lipid profile of rats across diverse groups was observed because of administration of high-fat diet (Figure 9). The PC group exhibited elevated levels of total cholesterol, triglycerides and low-density lipids. On the other hand, both the standard and treatment groups demonstrated a significant reduction in these lipid levels at the end of the experimental trial. In contrast, high-density lipids were markedly increased in the standard and AVM groups than in the PC group. This suggests the potential of *Artemisia vulgaris* in lowering lipid profile.

### 2.7. Serum Myocardial Enzymes and Natriuretic Peptides

Following disease induction, the levels of serum creatine kinase-MB, lactate dehydrogenase, atrial natriuretic peptide and B-type natriuretic peptide were markedly elevated due to cardiac muscle damage, and this elevation was sustained throughout the trial in PC group. Both the standard and *Artemisia vulgaris* treatments significantly reduced these levels, while *Artemisia vulgaris* more effectively maintained these markers toward normalcy according to the trial results, which shows its cardio protective role (Figure 10).

### 2.8. Serum Electrolytes

Serum electrolytes, including sodium and calcium, exhibited a notable increase in the PC group, while serum potassium significantly decreased in the same group due to variation in heart rate. On the other hand, the standard (STD) and *Artemisia vulgaris* methanolic (AVM) groups displayed the opposite trend, indicating normal heart contractions (Figure 11).

### 2.9. Inflammatory Markers

Notably, elevated levels of serum C-reactive protein (CRP) and tumor necrosis factor-alpha (TNF-α) were observed in the PC group due to its diseased condition. Treatment with standard metformin and *Artemisia vulgaris* significantly reduced the levels of these inflammatory markers, showing its anti-inflammatory potential (Figure 12). 

### 2.10. Pancreatic Histopathology

Figure 13 shows photomicrographs of pancreatic tissue from rats in different groups. A is the negative control showing normal histological features of islets of Langerhans and scattered beta cells visible in the vicinity, B is the positive control group with low beta cell numbers due to selective destruction of islets of Langerhans, C is the standard control group showing preserved histological features of islets of Langerhans, and D is the *Artemisia vulgaris*-treated group showing an increased number of beta cells indicating the regenerative property of *Artemisia vulgaris.* The changes in the diameter and area of the islets of Langerhans in the different groups are shown in Figure 14 and Appendix A.

### 2.11. Heart Histopathology

Figure 15 shows photomicrographs of myocardial tissue from rats in different groups. Figure 15A shows the normal histological features of the negative control group, B shows the deranged structure, edema in the myocardial muscles, hypertrophy and prominent muscle fiber breakage of the positive control group, C shows the mild disarray of muscle fibers in the cardiac tissue of the standard control group, and D shows the nearly normal structure of myocardial muscle in the *Artemisia vulgaris*-treated group.

### 2.12. Body Weight and Electrocardiography (ECG)

Before induction, i.e., until 8 weeks, body weight increased in the high-fat diet group, but after induction, body weight decreased in the that group. Standard and *Artemisia vulgaris* treatment restored the body weight to normal at the end of the trial. The overall relative organ weight was significantly lower (*p* ≤ 0.05) in both the standard and *Artemisia vulgaris* groups than in the PC group, as illustrated in Appendix A.

Electrocardiograms of the different groups of rats are shown in Figure 16. In the case of the negative control (NC), the ECG profiles revealed a euvital state characterized by a normal heart rate and unaltered intervals, while the positive control (PC) exhibited severe bradycardia, along with a broad QRS complex and prolonged PR, RR and QT intervals, which indicated decreased contraction of the heart muscles. In the standard control (STD) group, improvements in cardiac function were observed, including an enhanced heart rate and shortening of the QRS, PR, QT and RR intervals. In the *Artemisia vulgaris* treatment group (AVM), marked enhancements in heart rate were evident, alongside even intervals and narrowing of the QRS complex, with significant reductions in the PR, QT and RR intervals, which show increased force of heart contraction to nearly normal levels, converging toward physiological norms (Figure 17).

## 3. Discussion

Diabetic cardiomyopathy (DCM), a multifaceted condition marked by elevated glucose, insulin, HbA1c, oxidative stress, inflammation, lipid levels, electrolyte imbalance, left ventricular hypertrophy, fibrosis, cardiac muscle damage, elevated myocardial enzymes and natriuretic peptides, which are also indicators of heart damage. This cardiac damage impacts kidney and liver function by deteriorating liver function enzymes and renal function markers [8,24]. All analytical parameters reveal distinct aspects and are closely correlated with each other, i.e., alpha-amylase and alpha-glucosidase inhibition manage hyperglycemia; oxidative stress markers indicate both cellular damage and antioxidant defense, crucial in diabetic heart issues; and lipid profiles, electrolytes and enzymes reveal metabolic and heart health insights. Our study investigated *Artemisia vulgaris* effects on DCM and showed that *Artemisia vulgaris* significantly normalized all these parameters, attributed to its phytochemical constituents identified through comprehensive phytochemical analysis. Phenols and flavonoids are known for their potent antioxidant properties that counteract free radicals [25]. These compounds exhibit strong antioxidant activity, contributing to their ability to effectively inhibit key enzymes like alpha-amylase and alpha-glucosidase, pivotal in diabetes [26]. The greater the antioxidant potential, the greater the inhibition of alpha-amylase and alpha-glucosidase. This dual mechanism highlights the therapeutic potential of phenols and flavonoids in managing diabetes and its complication associated with inflammation and oxidative stress [27].

In our study, the methanolic extract of *Artemisia vulgaris* displayed significant total phenolic and flavonoid content 24.03 ± 2.64 (mg/g GAE) and 16.16 ± 0.72 (mg/g QE), respectively, while the HPLC analysis verified that phenolic compounds with P-coumaric acid in abundance, which has antifibrotic effects due to the inhibition of α-SMA [28]. P-coumaric acid also has a role in reducing blood glucose levels due to its anti-hyperglycemic effects via inhibiting α-amylase and α-glucosidase enzymes and cures diabetes [29,30]. 

Two enzymes, α-amylase and α-glucosidase, play important roles in carbohydrate digestion as well as postprandial blood glucose levels, indicating that their inhibition is valuable for treating diabetes-related cardiovascular complications [31]. Flavonoids, such as gallic acid, found in *Artemisia vulgaris* extract, inhibit α-amylase by inducing structural changes. Tannins found in plant extracts also play a role in α-amylase inhibition [32]. Chlorogenic acid, which is also present in plant extracts, inhibits both α-glucosidase and α-amylase [33]. Strong antidiabetic effects are shown by the significant decrease in blood glucose levels caused by the inhibition of these enzymes. The current study showed that the inhibitory effects of *Artemisia vulgaris* extract on α-amylase and α-glucosidase were similar to those of the standard drug acarbose. The presence of p-coumaric acid and chlorogenic acid in *Artemisia vulgaris* extract also contributed to these therapeutic effects, in accordance with previous research [34,35].

Prolonged high blood sugar levels in individuals with diabetes increase the risk of complications such as diabetic cardiomyopathy. Addressing insulin resistance and hyperinsulinemia is crucial for preventing diabetic cardiomyopathy [36]. *Artemisia vulgaris* mitigated hyperglycemia and restored serum insulin. Artemisinin is also found in *Artemisia vulgaris* and has been shown to improve type 2 diabetes as well as insulin resistance [37]. The HbA1c level is the average blood glucose level, and the HbA1c level remained high in the PC group; however, due to the decrease in blood and serum glucose levels in the STD and AVM groups, the HbA1c level also decreased in these groups. This is in accordance with previous research [37]. This suggests a potential shift from costly conventional antidiabetic medications to more affordable natural products for type 2 diabetic patients.

Diabetes leads to increased oxidative stress and reactive oxygen species (ROS) which are associated with the development of DCM. Antioxidants can counteract this oxidative stress. *Artemisia vulgaris* extract has been shown to have significant antioxidant effects, and its high phenolic content contributes to this antioxidant effect by scavenging various radicals such as superoxide anions and hydroxy radicals [38]. We analyzed the antioxidant activity of the plants, and the DPPH-radical-scavenging activity was 78.33 ± 1.65% at 5 mg/mL. The *Artemisia vulgaris* treatment resulted in a decrease in total oxidative stress (TOS) as well as melanodialdehyde (MDA) levels and increased total antioxidant capacity (TAC). These observations are in line with previous research that underscores the antioxidant effect of *Artemisia vulgaris* due to the presence of diverse phenolic compounds and flavonoids [39]. GC–MS analysis revealed the presence of diverse compounds with anti-inflammatory, antioxidant, antidiabetic, antitumor, vasodilatory and cardioprotective properties that contribute to the treatment of diabetic cardiomyopathy. *Artemisia vulgaris* extract had the highest concentration of oleic acid. One of its noteworthy attributes is its antioxidant ability because it can directly modulate the activation and synthesis of antioxidant enzymes [15].

Diabetes and its associated pathological conditions are known to strongly depend on the liver for glucose regulation. Hyperglycemia noticeably elevates liver enzymes, with a significant increase (50% to 80%) in AST and ALT levels. *Artemisia vulgaris* also demonstrates hepatoprotective properties through secondary metabolites of flavonoids, terpenes and phenolic acids [40]. Diabetes causes damage to kidneys and results in increased levels of urea and creatinine in the PC group, while the STD and AVM groups showed marked decreases in urea and creatinine, indicating the protective role of *Artemisia vulgaris* against diabetes-induced damage, in line with previous research [41]. In terms of cardiovascular health, *Artemisia vulgaris* shows hypolipidemic, anti-inflammatory and antioxidant properties, leading to reduced cholesterol and triglyceride levels and improved HDL levels. The phenolic components in plants delay lipid peroxidation by scavenging free radicals [42]. Within *Artemisia vulgaris*, the presence of P-coumaric acid inhibits HMG-CoA reductase, leading to a reduction in cholesterol biosynthesis in the liver and the restoration of imbalanced lipid levels to normal [43]. Furthermore, research has shown that compared with rosuvastatin, *Artemisia vulgaris* extract decreases serum lipid levels, offering potential benefits for individuals with hyperlipidemia [13]. Elevated levels of the cardiac markers LDH and CK-MB are observed in cardiac injury, a condition observed in diabetic cardiomyopathy. A study demonstrated that cinnamon, a natural product, decreased the levels of these enzymes [44]. These findings are in line with our research in which *Artemisia vulgaris* treatment of diabetic rats significantly reduced the expression of these markers to normal levels. Quercetin and Chlorogrnic acid, due to its ACE inhibitory and antioxidant properties, decreases the levels of cardiac biomarkers such as CK-MB and LDH, providing cardio protection [45,46]. Natriuretic peptides (ANP and BNP) serve as heart failure biomarkers. In diabetic rats, ANP and BNP levels are high [47,48], but *Artemisia vulgaris* treatment notably decreases these levels. Phenols and flavonoids maintain heart health by reducing oxidative stress and inflammation, preventing excessive release of natriuretic peptide and promoting cardiac balance [49].

Elevated inflammatory markers like CRP and TNF-α levels were observed in diabetic cardiomyopathy patients in line with our study [50]. Increased levels of CRP, an inflammatory marker in diabetes, exacerbate insulin resistance and diabetic cardiomyopathy through chronic inflammation. Sinapic acid present in *Artemisia vulgaris* extract regulates NF-κB signaling pathways, reducing inflammation and improving myocardial function [51]. TNF-α, another pro-inflammatory factor in diabetes, leads to cardiac dysfunction, oxidative stress and insulin resistance. 3-Methyl-2-(2-oxopropyl)furan and gallic acid present in *Artemisia vulgaris* extract reduce inflammation and inflammatory marker levels [17,52].

Histopathological analysis demonstrated that *Artemisia vulgaris* treatment restored diabetic rat pancreatic β-cells and the myocardium. A remarkable restoration of the endocrine portion, particularly β-cells, was observed. This increase in β-cells suggested regeneration, which might be due to the presence of quercetin in the plant extract [53]. STZ damages pancreatic β-cells through the generation of ROS, elevation in cytosolic calcium levels and reduction in antioxidant enzyme activities [54]. Antioxidants in *Artemisia vulgaris* reduce oxidative stress, thereby restoring β-cells. Myocardium histology revealed hypertrophy, myocardial degeneration and muscle breakage, but *Artemisia vulgaris* treatment reversed these changes. This effect may be attributed to kaempferol, which inhibits cyclooxygenase enzymes and reduces inflammation [55], as well as chlorogenic acid, which mitigates cardiac fibrosis via the nitric oxide/cGMP pathway [56]. *Artemisia vulgaris* extract contains gallic acid, which is known to reduce the inflammatory response by reducing the release of inflammatory cytokines, adhesion molecules, cell infiltration and chemokines [52].

Cardiac dysfunction can occur rapidly, with changes occurring in as little as 15 days. Diabetes leads to irregular diastolic and systolic pressure regulation, involving sodium and calcium retention and potassium loss which causes disturbance in heart rate and ECG; [57] in line with our findings, *Artemisia vulgaris* significantly restored serum electrolytes and ECG. Irregular hemodynamics play a part in hypertrophic cardiomyopathy and fibrosis, leading to ECG abnormalities. Treatment with *Artemisia vulgaris* significantly reversed these changes, possibly due to the presence of 1,25-dihydroxycholecalciferol, which is cardioprotective and antifibrotic [58]. Abnormal factors such as endothelial dysfunction, systemic inflammation, elevated free fatty acids and vascular issues related to advanced glycation are linked with ECG abnormalities [59]. Another study by Wang and team confirmed that streptozotocin-induced DCM resulted in abnormal ECG changes, such as prolonged QRS, PR and QT intervals with decreased heart rate. In that study, piperine, a natural substance, was employed because of its effectiveness in improving streptozotocin-induced diabetic cardiomyopathy and myocardial derangement and restoring the ECG intervals to almost normal levels [57]. Similarly, our study used a natural product, *Artemisia vulgaris*, which effectively reversed cardiac derangements and abnormalities on ECG. Another study [43] highlighted the protective effects of p-coumaric acid against cardiac hypertrophy, accompanied by the near normalization of ECG. Notably, *Artemisia vulgaris*, the treatment in our study, is rich in p-coumaric acid, suggesting a potential mechanism for the observed reversal of cardiac changes and ECG abnormalities. The cardioprotective effects induced by medicinal plants primarily involve increased antioxidant activity, resulting in the mitigation of ROS production. Additionally, these plants inhibit inflammatory signaling pathways and related cytokines. Furthermore, they enhance the function of the Na+/K+ ATPase pump, regulate the L-type Ca2+ channel current and influence intracellular ATP levels. The downregulation of TGFβ1 and TNFα expression further contributes to these protective effects [60]. Given these potential mechanisms, it is plausible that *Artemisia vulgaris*, a medicinal plant, may alleviate diabetic cardiomyopathy. An eminent facet of our study was the efficacy of *Artemisia vulgaris* in myocardial protection and diabetes management. Nonetheless, this research encountered certain limitations, including the short duration of the trail in ascertaining enduring cardiac functional enhancements and the exclusive exploration of a singular oral administration route. Future endeavors should encompass comprehensive preclinical and clinical investigations of *Artemisia vulgaris* across various administration routes.

## 4. Materials and Methods

### 4.1. Chemicals and Kits

Except as indicated, all chemicals were purchased from Sigma-Aldrich^®^, St. Louis, MO, USA. All other routine chemicals were obtained from a legitimate local chemical supplier. This study used streptozotocin from Sigma-Aldrich^®^, St. Louis, MO, USA (CAS-No: 18883-66-4), metformin from Martin Dow^®^, Darmstadt, Germany, Alpha amylase from Sigma-Aldrich^®^, St. Louis, MO, USA (Lot # BCBJ8997V), 2,2-Diphenyl-1-picrylhydrazyl from Alfa Aesar, Lancashire, UK (Lot #U07B055), and alpha glucosidase from Sigma-Aldrich^®^, St. Louis, MO, USA (CAS-No 9033-06-1). Each assay lists a catalog number for the kits used. We obtained high purity for all the chemicals used.

### 4.2. Experimental Animal Models and Experimental Conditions

A total of 27 albino rats were purchased and housed in the animal house of the Department of Physiology, Government College University, Faisalabad (GCUF). The average weight of the animals was 58 ± 10 g, and the average age was approximately 2 weeks. The rats were acclimatized to the animal experimental station under standard climatic conditions of 25 ± 5 °C and suitable humidity, i.e., 50 ± 5 %. There was a 12 h cycle of day and night lighting. Initially, the rats were divided into two main groups: the negative control (NC) group (n = 6), which received a normal diet and water, and the high-fat diet group (HFD) (n = 21), which received water with 5% sucrose and high-fat feed containing a cafeteria diet rich in cheese, potato chips, peanuts, high sugar biscuits and different kinds of diets containing high amounts of fats that mimic human dietary habits for type 2 diabetes development. According to previously published protocols [61], after two months of diet administration, when the weight of the animals reached 200–220 g, two doses of streptozotocin (STZ) (35 mg/kg bw) and nicotinamide (110 mg/kg bw) at 15 min intervals in 0.1 M freshly prepared citrate buffer were administered intraperitoneally to overnight-fasted rats. The second dose of STZ was administered after 3 days of the 1st dose. Fifteen days after induction, three rats were decapitated via cervical dislocation under anesthesia to avoid pain and to ensure the presence of diabetic cardiomyopathy (DCM) with elevated blood glucose levels and serum myocardial enzymes, and a histologically altered myocardial structure. Following the confirmation of DCM, rats receiving a HFD were divided into three groups: the positive control (PC) group (n = 6, standard (STD) group; n = 6, treated with metformin at a dosage of 200 mg/kg/day) and the treatment (AVM) group (n = 6, treated with the methanolic extract of *Artemisia vulgaris* at a dose of 300 mg/kg/day throughout the trial) [62]. The HFD group continued to progress to all groups except the NC group throughout the trial. Decapitation was carried out on day 45, and blood was drawn into EDTA vials for hematological analysis and into serum vials for biochemical analysis. The pancreas and heart were collected in 10% formaldehyde for histological examination.

### 4.3. Extract Preparation

The *Artemisia vulgaris* was purchased from Faisalabad, Pakistan, and was identified by an expert botanist from GCUF (Voucher No. 405-bot-24). The roots of the plant were cleaned and sun-dried. The air-dried roots were then ground into powder. Five hundred grams of the powder was soaked in 1500 mL of methanol for 72 h with continuous agitation, filtered, and evaporated using a rotary evaporator (SCILOGEX; Model RE100-Pro) at 55 °C, 0.06 MPa and 36 RPM. The obtained concentrated solution was poured into a Petri dish, incubated at 40 °C overnight and then dried to remove any residual methanol [12]. 

### 4.4. In Vitro Qualitative Phytochemical Analysis of Artemisia vulgaris

To check the presence of carbohydrates, the plant extracts were dissolved in distilled water and filtered to obtain clear solutions. These were mixed with equal amounts of Benedict’s reagent and gently heated, resulting in orange–red precipitates, indicating the presence of reducing sugars. For proteins, the extract was mixed with concentrated nitric acid, causing a yellow coloration, indicating the presence of proteins. For alkaloids, extract was dissolved in dilute HCl and filtered, which was then mixed with Hager’s reagent, producing a yellow precipitate, indicating the presence of alkaloids. For steroids and terpenoids, the extract was mixed with chloroform and filtered. The addition of a few drops of H_2_SO_4_ and shaking produced a red color in the lower layer, indicating steroids. A reddish-brown color resulting without shaking indicated the presence of terpenoids. To check the presence of phenols and tannins, the extract was mixed with a 10% lead acetate solution, resulting in the formation of white precipitates, indicating the presence of phenolics and tannins. The presence of flavonoid was checked when the plant extract was treated with NaOH solution, followed by dilute acetic acid; an intense yellow coloration formed which then faded to a colorless state, indicating the presence of flavonoids [63]. 

### 4.5. Determination of TPC, TFC and DPPH

The total phenolic content (TPC) was measured by adding 30 µL of plant extract (at a concentration of 1 mg/mL) with 100 µL of Folin–Ciocalteau reagent and 200 µL of 2.5% Na_2_CO_3_. After 60 min of incubation at room temperature, the absorbance was taken at 760 nm by using gallic acid as a reference standard. To determine total flavonoid content (TFC), 100 µL of plant extract (1 mg/mL) was mixed with 1 mL of distilled water. After a 5 min incubation at room temperature, 125 µL of AlCl_3_ and 75 µL of 5% NaNO_2_ were added, and the mixture was incubated for 6 min at room temperature. To conclude, 125 µL of 1 M NaOH was added, and the final volume was adjusted to 2.5 mL with distilled water. The absorbance was measured at 540 nm, and quercetin was used as a reference standard. To estimate antioxidant activity, sample solutions of five different concentrations (5, 2.5, 1.25, 0.62 and 0.312 mg/mL) were prepared in methanol, and each sample solution (5 μL) was mixed with 585 μL of a 0.2% DPPH (2,2-Diphenyl-2-picrylhydrazyl) solution prepared in methanol. This mixture was incubated for 20 min at room temperature, and the absorbance was taken at 515 nm. The higher the scavenging percentage, the greater the antioxidant activity, indicating the ability of the sample to neutralize free radicals. The radical-scavenging potential was calculated by the given formula [27]:$$\mathrm{Scavenging}\;(\%)=\frac{Abs.\;of\;blank-Abs.\;of\;sample}{Abs.\;of\;blank}\times 100$$

### 4.6. HPLC Analysis of Dry Extract

Dry extract, having been dissolved in methanol to achieve a concentration of 250 g/mL, underwent analytical scrutiny employing an HPLC system. Specifically, the HPLC system utilized was the Perkin Elmer^®^ instrument from the Waltham, Massachusetts United States. This analytical setup featured a Flexer Binary LC pump, a UV/VIS LC Detector, and a reverse-phase C18 column with dimensions of 5 mm in particle size and 4.6 mm in diameter. The analytical oven was maintained at a constant temperature of 30 °C throughout the analysis. The acquired data were subjected to meticulous analysis using Chromera software, version 4.1.2.6410.

The elution process was executed using a gradient technique, and the mobile phase comprised a mixture of 1% orthophosphoric acid in milli-Q water (A) and methanol (B). The eluent was delivered at a flow rate of 0.8 mL/min, and the temperature of the system was maintained at 30 °C. A sample volume of 10 μL was injected into the system for analysis. Notably, the UV spectrum of the samples was scrutinized within the wavelength range of 190 to 400 nm. Under the following conditions, an HPLC analysis was performed: injection volume, 10 μL; run time, 10 min; flow rate, 0.8 mL/min; temperature, 30 °C; and detection wavelength, 275 nm [64].

### 4.7. GC–MS of the Plant Extract

Plant extract underwent analysis utilizing GC-MS Agilent, Model 7890B (Santa Clara, CA, USA), coupled with Mass Hunter acquisition software (version 10.0.368) (Data APEX Ltd, Bellefonte, PA, USA, Pague 5). The instrumentation featured an ultra-inert capillary non-polar column (HP-5MS) with dimensions of 30 mm × 0.25 mm ID × 0.25 µm film thickness. A 1 µL sample was injected in split mode (ratio, 1:100), utilizing an electron ionization system operating at an ionization energy of 70 eV. Helium served as the carrier gas at a flow rate of 1.0 mL/min. The oven temperature was initially set at 50 °C for 5 min, then gradually increased to 250 °C at a rate of 100 °C/min, and finally to 300 °C for a duration of 10 min at a rate of 70 °C/min. Identification of metabolites present in the extract sample was performed using the NIST library [65].

### 4.8. α-Amylase and α-Glucosidase Inhibition Assay

To determine α-Amylase, various concentrations of the sample and standard acarbose both at concentrations of 5, 2.5, 1.25, 0.625 and 0.312 mg/mL were diluted. Then, 500 µL of each dilution was mixed with an equal volume of porcine pancreatic α-amylase solution (0.5 mg/mL) in a buffer system of 0.02 M sodium phosphate (pH 6.9) and 0.006 M NaCl. The reaction started at room temperature and proceeded for 10 min after adding 1% starch in 0.02 M PBS. The reaction was stopped with DNSA reagent after another 10 min incubation. The mixture was heated in a boiling water bath for 10 min and diluted with 10 mL of deionized water, and the absorbance was measured at 540 nm. To determine α-glucosidase, a solution was prepared by combining 980 μL of a PNPG mixture containing 290 mMole of β-D glucopyranoside in a 20 mMole citrate buffer (pH 5.6). To assess inhibitory potential, this solution was mixed with 200 µL of plant extract samples and standard acarbose, each at concentrations of 5, 2.5, 1.25, 0.625 and 0.312 mg/mL. After incubating the mixture at 37 °C for 5 min, 20 μL of a 1 U/mL α-glucosidase solution was added. The reaction proceeded for an additional 40 min at 35 °C. To stop the enzymatic reaction, 200 µL of 6N HCl was added. The absorbance of the resulting solution was measured at 405 nm. For each assay, a blank control was conducted without the addition of plant extract or acarbose to serve as a baseline reference for the enzyme’s full activity. The percentage enzyme inhibition was measured by the given formula [27]:$$\mathrm{Inhibition}\;(\%)\;=\frac{Abs.\;of\;blank-Abs.\;of\;sample}{Abs.\;of\;blank}\times 100$$

### 4.9. Determination of Glycemic Index

Rat fasting blood glucose levels (mg/dL) were measured weekly using a portable glucometer (ON-Call 609), and glycated hemoglobin (HbA1c; %) was measured by immunofluorescence using a GP Getein kit catalog # IF-1017. The serum glucose concentration (mg/dL) was determined using a glucose kit (Glucose-Liquizyme GOD-POD # 1422/3). The serum insulin concentration (ng/mL) was determined with an enzyme-linked immunosorbent assay (ELISA) using a serum insulin kit from GP Getein catalog # F-280.

### 4.10. Measurement of Oxidative Stress Parameters

The procedure mentioned in the study of Anwar and team was used to measure total antioxidant capacity (TAC; mmol Trolox equiv./L) and total oxidant status (TOS; μmol H_2_O_2_ equiv./L) with a BIOLAB-310 spectrophotometer, while malondialdehyde (MDA; mmol/L) was determined by using a HITACHI-UHS300 spectrophotometer [64].

### 4.11. Determination of Serum Liver Enzymes

A commercially available kit supplied by Innoline, catalog # A220073, was used to quantify aspartate aminotransferase (AST; U/L) in the serum via a calorimetric method. The serum alanine aminotransferase (ALT; U/L) concentration was also determined by an Innoline kit (catalog # A220432) via a calorimetric method.

### 4.12. Determination of Renal Function Markers

The serum creatinine (mg/dL) concentration was quantified by a calorimetric method using a kit from innoline (catalog # A220647), and the serum urea (mg/dL) concentration was also measured by a calorimetric method from an innoline kit (catalog # A220172).

### 4.13. Determination of Lipid Profiles

The lipid profile was measured by a calorimetric method using a spectrophotometer. Total cholesterol (TC; mg/dL) was quantified using the cholesterol Lab kit, catalog # T-7764. Triglycerides (TG; mg/dL) were measured with the Tersaco Switzerland LIQ-488B Kit. The high-density and low-density lipoprotein-cholesterol (HDL-LDL; mg/dL) were measured by the Diasys Diagnostic System catalog # 60,147,058 and catalog # 60145976, respectively.

### 4.14. Determination of Serum Myocardial Enzymes

A biochemistry analyzer (Biosystem BTS-330) was used to measure myocardial enzymes via a calorimetric method. Creatine kinase–myocardial band levels (CK-MB; IU/L) were measured by using a kit from Merck catalog # LIQ626A. Serum lactate dehydrogenase (LDH; IU/L) was determined spectrophotometrically by using Lab kit catalog #T-7751.

### 4.15. Determination of Natriuretic Peptide Levels

The levels of serum atrial natriuretic peptide (ANP; pg/mL) were measured with an enzyme-linked immunosorbent assay (ELISA) using the E-Lab Science Kit Catalog # E-CL-R0544. Serum levels of B-type natriuretic peptide (BNP; pg/mL) were measured by a competitive immunoassay from the GP Getein kit catalog # IF-1002.

### 4.16. Determination of Serum Electrolytes

Serum electrolytes such as sodium, potassium and calcium (mmol/L) were measured via the ion selective electrode method (ISE) by using the Diamond Diagnostics Catalog # IL-2121D.

### 4.17. Determination of Inflammatory Markers

Serum tumor necrosis factor-alpha (TNF-α; pg/mL) was measured by sandwich ELISA from the E-Lab Science Catalog # E-EL-R2856. Serum C-reactive protein (CRP; pg/mL) was also measured by a sandwich ELISA kit from Labway Diagnostics Catalog #LDLA003-100.

### 4.18. Histopathology Examination

Pancreatic and heart tissues were collected and immersed in 10% formaldehyde solution for 24 h to achieve proper fixation. Afterward, the tissues were dehydrated using different ethanol concentrations. Following dehydration, the tissues were encased in paraffin wax and precision-cut into sections measuring 2 to 4 micrometers in thickness using a microtome. These deparaffinized sections were then subjected to eosin and hematoxylin staining and subsequently scrutinized under a light microscope for analysis and evaluation.

### 4.19. Electrocardiography (ECG) and Body Weight

ECG recordings for each group were acquired using the PowerLab 15T system (ML4818) [64]. The body weight of each group was measured weekly throughout the trial, and the heart weight was measured on day 45 after decapitation.

### 4.20. Statistical Analysis

The sample data were entered and analyzed using GraphPad Prism version 9.5 and CoStat 2 software. A one-way analysis of variance (ANOVA) was employed, where statistical significance was defined as a *p* value ≤ 0.05. The data are expressed as the mean ± SEM.

## 5. Conclusions

Based on our findings, it is concluded that *Artemisia vulgaris* roots exerts overall beneficial effects primarily through its phytochemical compounds, conferring anti-inflammatory, cardioprotective, antioxidant, antidiabetic and antidyslipidemic properties. Along with observed histological prevention of cardiac hypertrophy and myocardial derangement, this study establishes, for the first time, the potential bioactivity of *Artemisia vulgaris* roots against diabetic cardiomyopathy through the maintenance of healthy blood chemistry and the preservation of cardiac integrity. The findings strongly advocate for comprehensive preclinical and clinical investigations in the future aimed at the development of nutraceuticals based on *Artemisia vulgaris* for DCM treatment.

## Figures and Tables

**Figure 1 pharmaceuticals-17-01046-f001:**
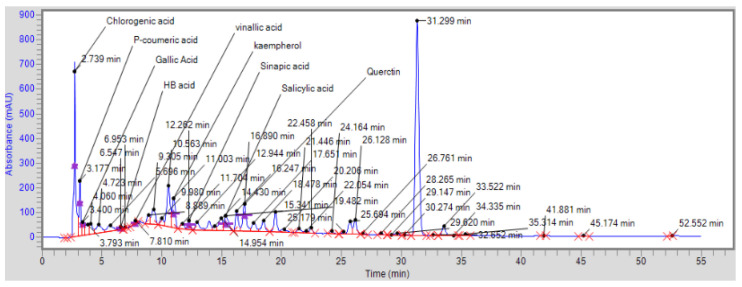
HPLC chromatogram of *Artemisia vulgaris*.

**Figure 2 pharmaceuticals-17-01046-f002:**
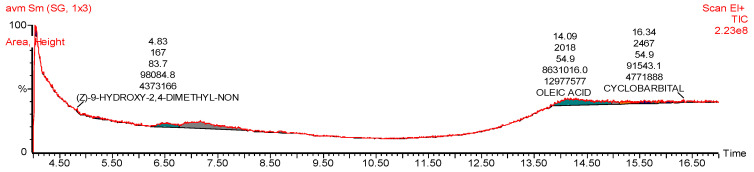
GC–MS chromatogram of *Artemisia vulgaris*.

**Figure 3 pharmaceuticals-17-01046-f003:**
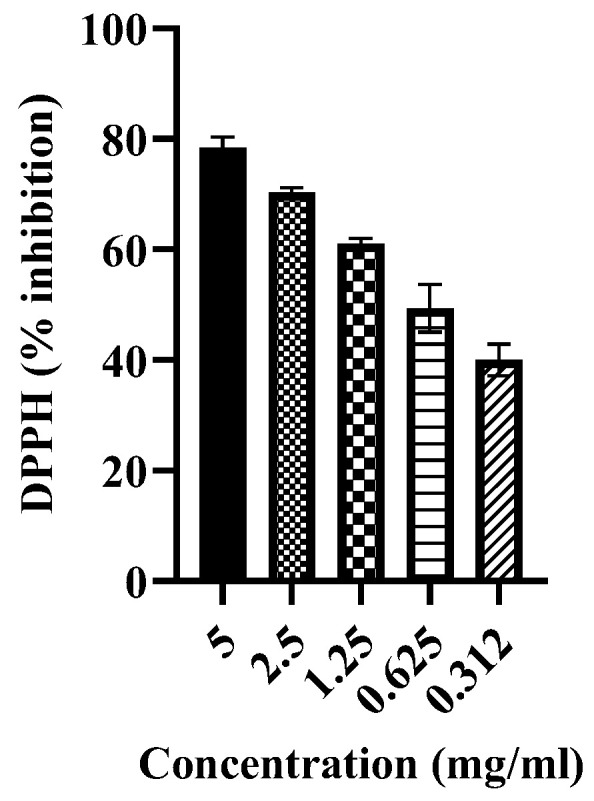
DPPH percentage inhibition at different concentrations of the *Artemisia vulgaris* methanolic extract. Different bars represent concentration gradient variations; higher concentrations correlate with greater percentage inhibition.

**Figure 4 pharmaceuticals-17-01046-f004:**
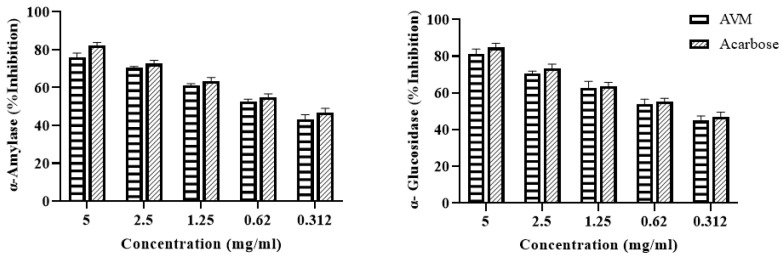
Percentage inhibition of α-amylase and α-glucosidase by different concentrations of *Artemisia vulgaris* methanolic extract (AVM) and standard acarbose.

**Figure 5 pharmaceuticals-17-01046-f005:**
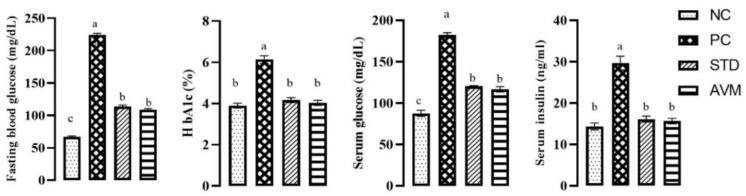
Glycemic indices of the negative control (NC), positive control (PC), standard (STD) and *Artemisia vulgaris* methanolic extract (AVM) groups. Different letters on the bars for each parameter indicate a statistically significant difference (*p* ≤ 0.05).

**Figure 6 pharmaceuticals-17-01046-f006:**
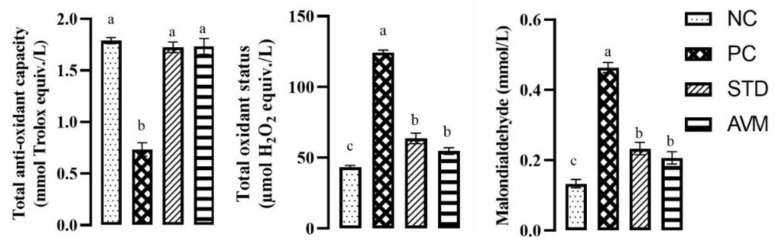
Oxidative stress parameters of the negative control (NC), positive control (PC), standard (STD) and AVM groups. Different letters on the bars for each parameter indicate a statistically significant difference (*p* ≤ 0.05).

**Figure 7 pharmaceuticals-17-01046-f007:**
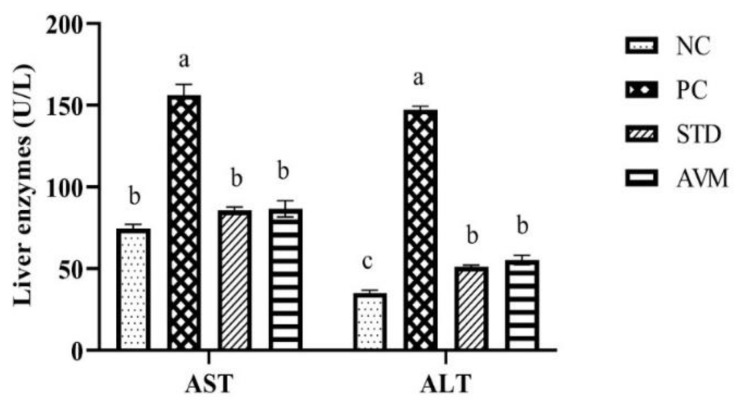
Serum liver enzymes (AST; aspartate amino transferase, ALT; alanine aminotransferase) in the negative control (NC), positive control (PC), standard (STD) and *Artemisia vulgaris* methanolic extract (AVM) groups. Different letters on the bars for each parameter indicate a statistically significant difference (*p* ≤ 0.05).

**Figure 8 pharmaceuticals-17-01046-f008:**
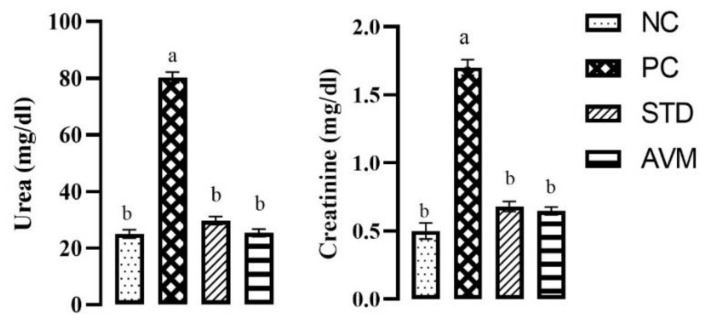
Renal function markers in the negative control (NC), positive control (PC), standard (STD) and AVM groups. Different letters on the bars for each parameter indicate a statistically significant difference (*p* ≤ 0.05).

**Figure 9 pharmaceuticals-17-01046-f009:**
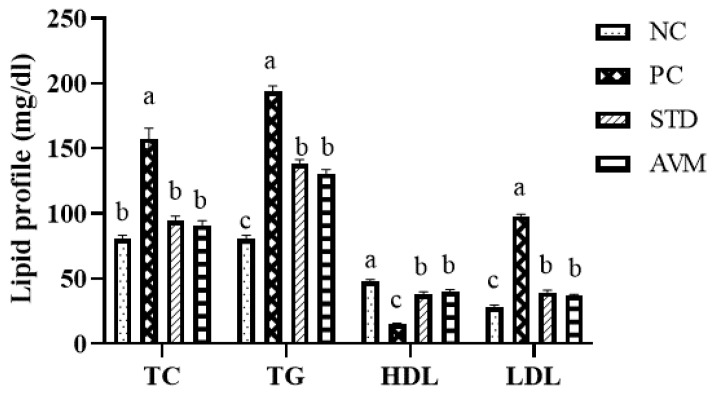
Lipid profile (TC; total cholesterol, TG; triglycerides, HDL; high-density lipids, LDL; low-density lipids) of the negative control (NC), positive control (PC), standard (STD) and AVM groups. Different letters on the bars for each parameter indicate a statistically significant difference (*p* ≤ 0.05).

**Figure 10 pharmaceuticals-17-01046-f010:**
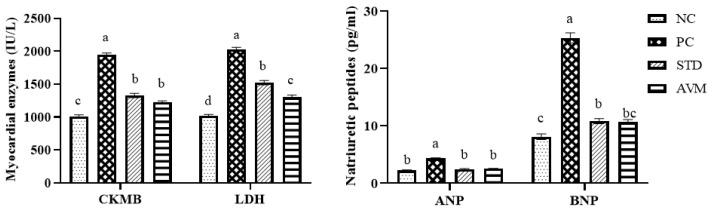
Serum myocardial enzymes (CK-MB; creatine kinase-MB, LDH; lactate dehydrogenase) and natriuretic peptides (ANP; atrial natriuretic peptide, BNP; B-type natriuretic peptide) in the negative control (NC), positive control (PC), standard (STD) and *Artemisia vulgaris* methanolic extract (AVM) groups. Different letters on the bars for each parameter indicate a statistically significant difference (*p* ≤ 0.05).

**Figure 11 pharmaceuticals-17-01046-f011:**
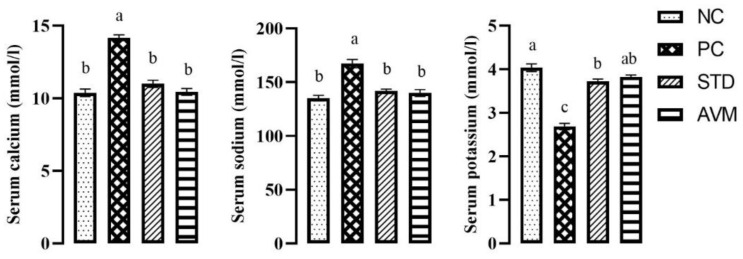
Serum electrolytes of negative control (NC), positive control (PC), standard (STD) and *Artemisia vulgaris* methanolic extract (AVM) groups. Different letters on the bars for each parameter show a statistically significant difference (*p* ≤ 0.05).

**Figure 12 pharmaceuticals-17-01046-f012:**
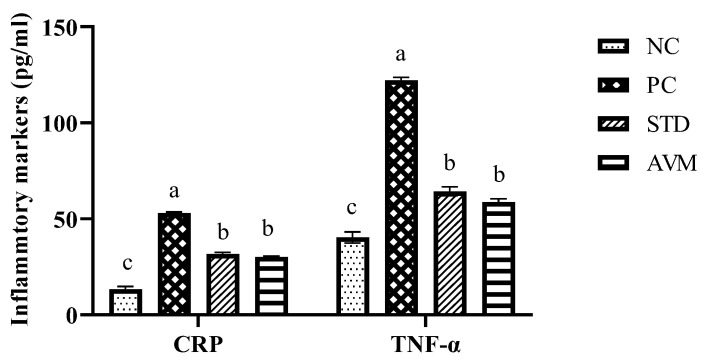
Inflammatory marker (CRP: C-reactive protein, TNF-α: tumor necrosis factor-alpha) levels in the negative control (NC), positive control (PC), standard (STD) and *Artemisia vulgaris* methanolic extract (AVM) groups. Different letters on the bars for each parameter indicate a statistically significant difference (*p* ≤ 0.05).

**Figure 13 pharmaceuticals-17-01046-f013:**
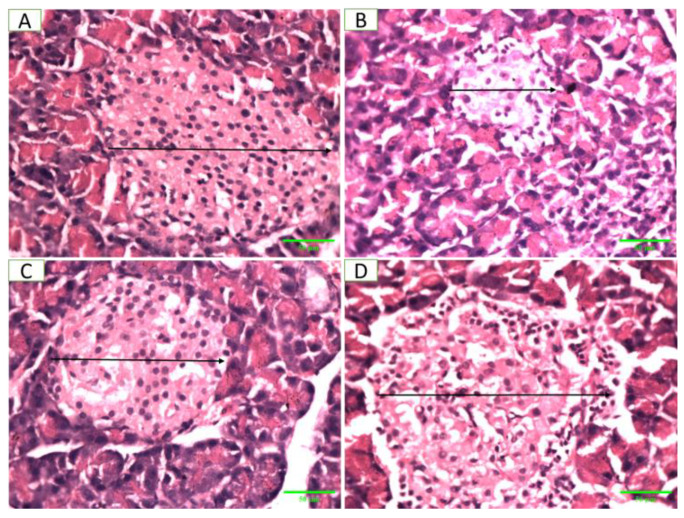
Photomicrographs of pancreatic tissue from different groups of rats with STZ-induced DCM. The arrow indicates the diameter of the islets of Langerhans (hematoxylin and eosin stain, 400×; scale bar, 50 µm).

**Figure 14 pharmaceuticals-17-01046-f014:**
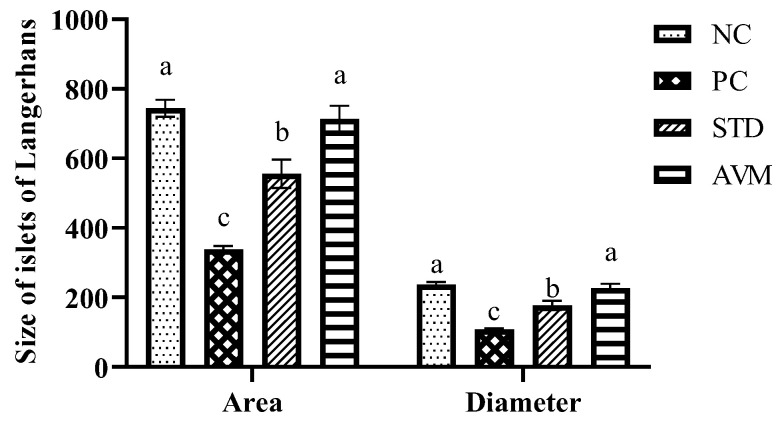
Size of islets of Langerhans (area: µm^2^, diameter: µm) in the negative control (NC), positive control (PC), standard (STD) and *Artemisia vulgaris* methanolic extract (AVM) groups. Different letters on the bars for each parameter indicate a statistically significant difference (*p* ≤ 0.05).

**Figure 15 pharmaceuticals-17-01046-f015:**
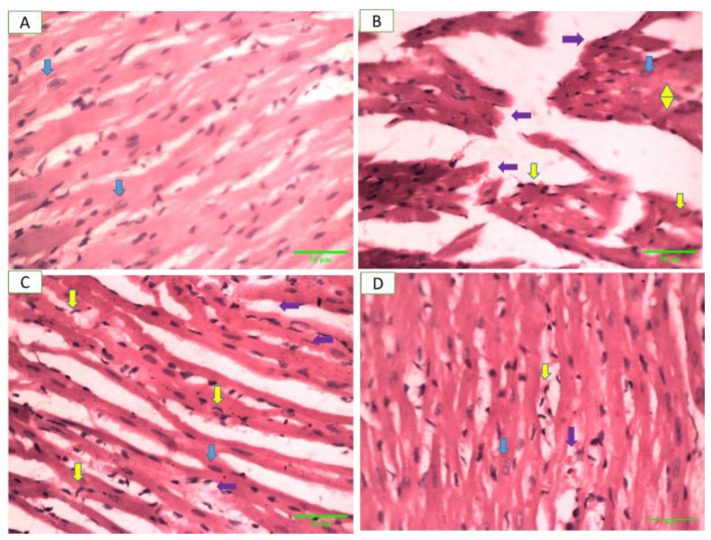
Photomicrographs of myocardial tissue from different groups of rats with STZ-induced DCM. Blue arrow: centrally located nuclei, yellow arrow: fibroblast, purple arrow: degeneration of cardiac myocytes, yellow double arrowhead: hypertrophy (H&E stain, 400×; scale bar, 50 µm).

**Figure 16 pharmaceuticals-17-01046-f016:**
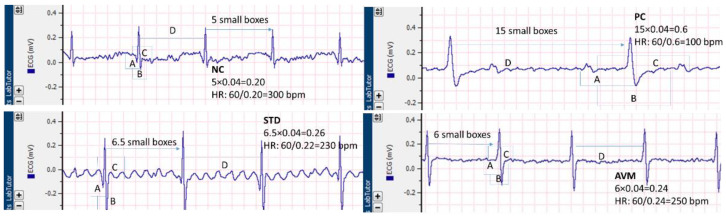
ECG recordings of different groups of rats in the STZ-induced DCM rat model. Labeled duration A = PR, B = QRS, C = QT and D = RR interval.

**Figure 17 pharmaceuticals-17-01046-f017:**
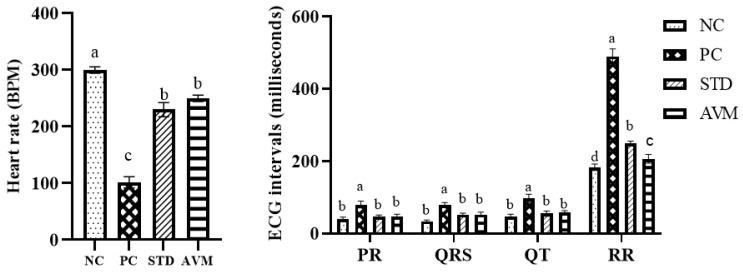
ECG parameters of the negative control (NC), positive control (PC), standard (STD) and AVM groups. Different letters on the bars for each parameter indicate a statistically significant difference (*p* ≤ 0.05). BPM: beats per minute.

**Table 1 pharmaceuticals-17-01046-t001:** Qualitative phytochemical analysis of *Artemisia vulgaris*.

Proteins	+	Steroids	++
Carbohydrates	++	Terpenoids	++
Alkaloids	+++	Flavonoids	++
Phenols	+++	Tannins	+++

**Table 2 pharmaceuticals-17-01046-t002:** Quantification of various phenolic and flavonoid compounds of *Artemisia vulgaris* through HPLC.

Sr. #	Name	Retention Time	Peak Area	K Factor	Concentration mg/kg of Dry Extract
1	Chlorogenic acid	2.739	6,084,547.8	0.00019	1156.06
2	P-coumaric acid	3.177	2,389,860.3	0.0003	7169.5
3	Gallic acid	3.400	717,613.9	0.000079	56.69
4	Vinallic acid	7.810	241,731.8	0.000044	10.63
5	Kaempherol	11.003	2,076,264.6	0.000321	666.4
6	Sinapic acid	12.262	561,109.5	0.00005	28.05
7	Salicylic acid	15.341	1,451,148.5	0.000377	547.08
8	Quercetin	16.890	2,260,742.5	0.000095	214.7

**Table 3 pharmaceuticals-17-01046-t003:** GC–MS analysis of *Artemisia vulgaris*.

Sr. #	Name	% Area	M.F	M.W	Pharmacological Activity
1	Oleic acid	35.74	C_18_H_34_O_2_	282.5 g/mol	Antidiabetic, Antioxidant, Anti-inflammatory, Anti-atherosclerosis, Anti-adhesive [14,15]
2	1,10Hexadecanediol	30.18	C_16_H_34_O_2_	258.44 g/mol	Antioxidant [16]
3	3-Methyl-2-(2-Oxopropyl) Furan	9.138	C_8_H_10_O_2_	138.16 g/mol	Anti-inflammatory, Antioxidant, Antimicrobial [17]
4	3-N-Hexylthiolane, S,S-Dioxide	9.238	C_10_H_20_O_2_S	204.33 g/mol	Antitumor, Anticancer [17]
5	7-Hexadecenal, (Z)-	0.853	C_16_H_30_O	238.41g/mol	Antioxidant, Anti-inflammatory [18]
6	Z,Z-6,28-Heptatriactontadien-2	1.458	C_37_H_70_O	530.9 g/mol	Vasodilator [19]
7	(Z)-9-Hydroxy-2,4-Dimethyl-Non	0.588	C_11_H_18_O_2_	182.26 g/mol	No activity reported
8	Cyclobarbital	0.600	C_12_H_16_N_2_O_3_	236.27 g/mol	CNS depressant [20]
9	1,25-Dihydroxyvitamin D3, TMS	3.363	C_30_H_52_O_3_Si	488.8 g/mol	Anti-inflammatory, Antioxidant, Cardioprotective, Antidiabetic [21,22]
10	Piperidine, 2-Propyl-, (S)-	0.7332	C_8_H_17_N	127.23 g/mol	Anticancer, Antiviral, Antimalarial, Analgesic, Anti-inflammatory, Anticoagulant [23]

M.F: Molecular formula, M.W: Molecular weight.

## Data Availability

Data presented in this study is available on request from the corresponding author.

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
