# Peer review of "Artemisia vulgaris* Extract as a Novel Therapeutic Approach for Reversing Diabetic Cardiomyopathy in a Rat Model"

_pharmaceuticals, 2024, doi:10.3390/ph17081046_

Round 1
Reviewer 1 Report
Comments and Suggestions for Authors
The study investigated the left ventricular remodeling potential of methanolic extract from Artemisia vulgaris in a rat model of diabetic cardiomyopathy (DCM). Researchers performed in vitro phytochemical analysis, including plant phenolic and flavonoid content determination, high-performance liquid chromatography (HPLC), and gas chromatography-mass spectrometry (GC-MS). The extract exhibited antioxidant activity and antidiabetic potential, reducing fasting blood glucose, glycated hemoglobin (HbA1c), serum glucose, insulin, oxidative stress markers, and more. ECG and histopathology confirmed significant improvement in heart remodeling. Artemisia vulgaris shows promise for DCM treatment due to its anti-inflammatory, cardioprotective, antidiabetic, and antidyslipidemic properties. Further research is warranted to explore its potential as a nutraceutical.
The following question should be addressed very carefully.
- What specific mechanisms are responsible for the observed cardioprotective effects of Artemisia vulgaris? Are there any molecular pathways involved?
- Have the researchers explored different dosages of the methanolic extract to determine the optimal therapeutic dose for DCM treatment?
- How does the treatment with Artemisia vulgaris impact long-term outcomes in DCM? Is there sustained improvement in cardiac function?
- What adverse effects, if any, were observed during the study? Is Artemisia vulgaris safe for prolonged use?
- How does Artemisia vulgaris compare to existing medications commonly used for DCM treatment, such as ACE inhibitors or beta-blockers?
- Are there any ongoing or planned clinical trials to validate the efficacy of Artemisia vulgaris in humans with DCM?
- Has the bioavailability of the methanolic extract been assessed? How well does it reach cardiac tissue?
- What impact does Artemisia vulgaris have on cardiomyocytes at the cellular level? Does it affect apoptosis, hypertrophy, or fibrosis?
- Are there any potential drug interactions when using Artemisia vulgaris alongside standard DCM medications?
- Did the study focus on a specific subset of diabetic patients (e.g., type 1 vs. type 2 diabetes)? How generalizable are the findings?
- Figure 17: Remove the software tagline from Figure 17.
- Reference Section: Ensure uniformity in page numbers across references. It is advisable to mention the significance of synthetic endeavors in the advancement of antidiabetic medication. Within this context, it is recommended to underscore the significance of iminosugars and sugar derivatives as potent antidiabetic agents. To support this assertion, referencing the subsequent pertinent articles in the introduction section is suggested: i) https://doi.org/10.1002/anie.202217809 ii) Compain, P.; Martin, O. R. Iminosugars: From synthesis to therapeutic applications; Wiley-VCH:New York, 2007; pp 187−298 and iii) https://doi.org/10.24820/ark.5550190.p011.809.
Author Response
Comment 1: What specific mechanisms are responsible for the observed cardioprotective effects of Artemisia vulgaris? Are there any molecular pathways involved?
Response 1: The pathways observed include, enhanced functioning of the Na+/K+ ATPase pump and regulation of L-type Ca2+ channel (Fig. 11 and Line 397), cardiac fibrosis attenuation via the nitric oxide/cGMP pathway (Page 12, Line 370), and via the inhibition of α-SMA (Page 11, Line 283). Decrease in lipotoxicity due to inhibition of HMG-CoA reductase (Page 12, Line 336). Reduction in oxidative stress via scavenging superoxide anions and hydroxy radicals effect (Page 11, Line 312). Direct synthesis of anti-oxidant enzyme (Page 11, Line 323). Improvement in myocardial function by reducing inflammation via regulation of NF-κB signaling pathway (Page 12, Line 355). Down regulation of TNF-α (Fig 12, Line 398). Inhibition of cyclooxygenase enzymes and reduction of inflammation (Page 12, Line, 369). Reduction of myocardial injury by decreasing myocardial enzyme through ACE inhibitor action (Page 12, Line 344).
Comment 2: Have the researchers explored different dosages of the methanolic extract to determine the optimal therapeutic dose for DCM treatment?
Response 2: We appreciate your question. The selection of the optimal dose for the methanolic extract of Artemisia vulgaris was informed by findings from literature sources, referenced as Gilani et al., 2005, which has been appropriately cited within our manuscript (Page 14, Line 439). This citation anchors our dosage selection in established research, ensuring the relevance and reliability of our experimental approach.
Comment 3: How does the treatment with Artemisia vulgaris impact long-term outcomes in DCM? Is there sustained improvement in cardiac function?
Response 3: Thank you for emphasizing this aspect. Our study did not extend the observation period adequately to assess sustained effects, which constitutes a limitation However, we aim to address this in future research endeavors (Page 13, Line 403).
Comment 4: What adverse effects, if any, were observed during the study? Is Artemisia vulgaris safe for prolonged use?
Response 4: No adverse effects were observed during the study. However, due to the limited duration of our investigation, we cannot definitively comment on the long-term safety of the intervention. However, existing literature suggests that Artemisia vulgaris is non-toxic even at higher doses like 3g/kg (Gilani et al., 2005) and is commonly used in culinary practices across Asia, where it is incorporated in soups and also used as a vegetable (Soon et al., 2019) (Page 2 and Line 71, 72).
Comment 5: How does Artemisia vulgaris compare to existing medications commonly used for DCM treatment, such as ACE inhibitors or beta-blockers
Response 5: Our treatment Artemisia vulgaris has detected Chlorogenic acid and quercetin, both of them are ACE inhibitors (Angiotensin converting enzyme inhibitor), as described in discussion too (Page 12, line 344) (Patel et al., 2018; Huang et al., 2017; Muhammad et al., 2015). On the other hand, if we talk about ACE inhibitors and beta blockers alone so they have many side effects such as ACE inhibitors can cause hyperkalemia, hypotension, kidney dysfunction, angioedema, altered taste, rash, and rare hematological reactions. Beta blockers may cause bradycardia, hypotension, fatigue, cold extremities, bronchoconstriction, masking of hypoglycemia symptoms, sleep disturbances, depression, sexual dysfunction, and rare serious effects like heart failure exacerbation and allergic reactions. Utilizing Artemisia vulgaris aims to achieve comparable therapeutic outcomes while minimizing these adverse effects associated with conventional existing medications.
Comment 6: Are there any ongoing or planned clinical trials to validate the efficacy of Artemisia vulgaris in humans with DCM?
Response 6: In our laboratory, our current research focus exclusively utilizes animal models for experimentation purposes. We do not conduct research involving human subjects. Our decision to focus on animal models stems from the need to explore biological mechanisms, test hypotheses, and investigate potential therapeutic interventions in a controlled and ethically sound environment. However, Future recommendations involve the planning of human trials. While our current research is centered on animal models (Page 13, Line 405 to 406) and (Page 17, Line 604, 605).
Comment 7: Has the bioavailability of the methanolic extract been assessed? How well does it reach cardiac tissue?
Response 7: This is an interesting question. We appreciate this question. Bioavailability is an established phenomena describing the proportion of a substance that reaches systemic circulation after oral administration. When ingested, substances enter the bloodstream via absorption in the gastrointestinal tract, circulating throughout the body, except certain physiological barriers. For instance, the heart receives a substantial blood flow, approximately 250-350 milliliters per minute in resting adults, ensuring widespread distribution of the drug to cardiac myocytes. However, this was not the aim of our study so we didn’t explore the bioavailability of extract. In future we will surely consider this.
Comment 8: What impact does Artemisia vulgaris have on cardiomyocytes at the cellular level? Does it affect apoptosis, hypertrophy, or fibrosis?
Response 8: Our in vivo model of diabetic cardiomyopathy initially focused on lipotoxicity induced by a high-fat diet, rather than on fibrotic or apoptotic structural changes. However, in the group supplemented with Artemisia vulgaris, significant restoration of cardiac myocyte damage and hypertrophy was observed, as depicted in Figure 15, Line 233.
Comment 9: Are there any potential drug interactions when using Artemisia vulgaris alongside standard DCM medications?
Response 9: This study involved two distinct treatment groups: one administered with a standard drug and the other supplemented with Artemisia vulgaris. These groups were separate entities, ensuring that the standard drug was not combined with the plant treatment, thereby eliminating the possibility of any drug interactions. By maintaining them as independent groups, we aimed to discern the individual effects of each treatment regimen on diabetic cardiomyopathy. This approach allowed us to evaluate Artemisia vulgaris's potential efficacy in comparison to established pharmaceutical therapies, providing a clear understanding of its therapeutic impact in mitigating diabetic cardiomyopathy.
Comment 10: Did the study focus on a specific subset of diabetic patients (e.g., type 1 vs. type 2 diabetes)? How generalizable are the findings?
Response 10: The protocols employed in this study were tailored for type 2 diabetes mellitus. Therefore, the findings and conclusions are specifically applicable to type 2 diabetes. The study did not investigate the effects of the extract on type 1 diabetes patients. This is mentioned in manuscript (Page 11, Line 306 to 307) and (Page 13, Line 426, 427).
Comment 11: Figure 17: Remove the software tagline from Figure 17.
Response 11: The suggested changing has been done (Page 10, Fig 16, Line 254).
Comment 12: Reference Section: Ensure uniformity in page numbers across references. It is advisable to mention the significance of synthetic endeavors in the advancement of antidiabetic medication. Within this context, it is recommended to underscore the significance of iminosugars and sugar derivatives as potent antidiabetic agents. To support this assertion, referencing the subsequent pertinent articles in the introduction section is suggested: i) https://doi.org/10.1002/anie.202217809 ii) Compain, P.; Martin, O. R. Iminosugars: From synthesis to therapeutic applications; Wiley-VCH:New York, 2007; pp 187−298 and iii) https://doi.org/10.24820/ark.5550190.p011.809.
Response 12: The suggested changes have been implemented, and the suggested reference has been added accordingly (Page 2, Line 66 to 69).
References
Gilani, Anwarul Hassan, Sheikh Yaeesh, Qamar Jamal, and M. Nabeel Ghayur. "Hepatoprotective activity of aqueous–methanol extract of Artemisia vulgaris." Phytotherapy Research: An International Journal Devoted to Pharmacological and Toxicological Evaluation of Natural Product Derivatives 19, no. 2 (2005): 170-172.
Soon, Laura, Phui Qi Ng, Jestin Chellian, Thiagarajan Madheswaran, Jithendra Panneerselvam, Gaurav Gupta, Srinivas Nammi et al. "Therapeutic potential of Artemisia vulgaris: An insight into underlying immunological mechanisms." Journal of Environmental Pathology, Toxicology and Oncology 38, no. 3 (2019).
Patel, Rahul V., Bhupendra M. Mistry, Surendra K. Shinde, Riyaz Syed, Vijay Singh, and Han-Seung Shin. "Therapeutic potential of quercetin as a cardiovascular agent." European journal of medicinal chemistry 155 (2018): 889-904.
Huang, Wu‐Yang, Lin Fu, Chun‐Yang Li, Li‐Ping Xu, Li‐Xia Zhang, and Wei‐Min Zhang. "Quercetin, hyperin, and chlorogenic acid improve endothelial function by antioxidant, antiinflammatory, and ACE inhibitory effects." Journal of Food Science 82, no. 5 (2017): 1239-1246.
Muhammad, Syed Aun, and Nighat Fatima. "In silico analysis and molecular docking studies of potential angiotensin-converting enzyme inhibitor using quercetin glycosides." Pharmacognosy magazine 11, no. Suppl 1 (2015): S123.
Reviewer 2 Report
Comments and Suggestions for Authors
Dear authors will find below the observations and comments made during the review and evaluation of their work. These must be taken into account when updating the manuscript.
1) The title is clear and informative, indicating the focus of the study on the therapeutic effects of Artemisia vulgaris on diabetic cardiomyopathy. It could be improved by specifying the method used, for example: "Artemisia Vulgaris Extract as a Novel Therapeutic Approach to Reverse Diabetic Cardiomyopathy in a Rat Model".
2) The abstract is comprehensive, describing the purpose of the study, methodology and key findings. However, from the abstract to the conclusions, the wording needs to be revised to express the ideas clearly and directly, so that the document can be easily read and understood.
3) Remove abbreviations from the abstract. Include them from the introduction to the end of the document.
4) All experimental protocols should be properly referenced and described in detail, not briefly. The methodology should open with details.
5) Some section of the document should indicate how all the analytical parameters that have been determined correlate, e.g. antioxidant capacity with a-glycosidase, glycaemic index, oxidative stress, enzymes, markers, lipid profile, electrolytes, etc. In addition, it should say why and for what purpose each test/method was performed and how the results correlate with each other.
6) Relevant information is missing in the methodology sections, e.g. in 2.5 for the UV/VIS dectector the wavelength used in the assays is missing, in 2.6 the operating conditions of the mass spectrometric detector are missing. Likewise, the criteria for comparison of the mass spectra obtained with the mass spectra in the Wiley database. It does not indicate the injection volume used and the injection mode whether it was Split or Splitless. As discussed above you should open each section of the methodology.
7) The discussion should follow a logical structure including: A) Summary of the main findings. B) Comparison with previous studies. C) Explanation of biological mechanisms. D) Clinical implications. E) Limitations of the study. F) Suggestions for future research.
8) Ensure that paragraphs are logically connected. Use transitional sentences to guide the reader from one point to the next.
9) Check that all statements are supported by appropriate citations and check that references are correctly formatted according to journal guidelines.
10) You should connect the introduction to the results: Make sure you clearly connect the findings discussed with the objectives stated in the introduction and with the results presented. This helps to reinforce the coherence of the manuscript.
11) Emphasise the key results of your study at the end of each subsection to remind the reader of the most important findings.
12) The end of the discussion should include a paragraph with the strengths and weaknesses of the work you have done and presented.
Authors must make changes and update your work
Regards
Reviewer
Comments on the Quality of English Language
Minor editing of English language required
Author Response
Comment 1: The title is clear and informative, indicating the focus of the study on the therapeutic effects of Artemisia vulgaris on diabetic cardiomyopathy. It could be improved by specifying the method used, for example: "Artemisia vulgaris Extract as a Novel Therapeutic Approach to Reverse Diabetic Cardiomyopathy in a Rat Model".
Response 1: The title has been revised as suggested (Page 1, Line 3).
Comment 2: The abstract is comprehensive, describing the purpose of the study, methodology and key findings. However, from the abstract to the conclusions, the wording needs to be revised to express the ideas clearly and directly, so that the document can be easily read and understood.
Response 2: The wording of abstract has been revised (Page 1, Line 11 to 14, 16 to 18 and 21 to 31).
Comment 3: Remove abbreviations from the abstract. Include them from the introduction to the end of the document
Response 3: The abbreviations have been eliminated from the abstract and appropriately included in the main manuscript, as advised (Page 1).
Comment 4: All experimental protocols should be properly referenced and described in detail, not briefly. The methodology should open with details.
Response 4: All experimental protocols are properly referenced and all experimental protocols have been added in details (from line no 452 to 532). I hope this significant addition can help this manuscript meet the standard of publication.
Comment 5: Some section of the document should indicate how all the analytical parameters that have been determined correlate, e.g. antioxidant capacity with a-glycosidase, glycaemic index, oxidative stress, enzymes, markers, lipid profile, electrolytes, etc. In addition, it should say why and for what purpose each test/method was performed and how the results correlate with each other.
Response 5: The significant aspects and correlation of analytical parameters has been incorporated in the manuscript (Page 10, Line, 266 to 270). The inhibition of alpha-amylase and alpha-glucosidase shows a direct correlation with antioxidant compounds, indicating a closely intertwined relationship between them. This correlation has been incorporated into the main text with reference (Page 10, Line no 277 to 279). In the investigation of diabetic cardiomyopathy, the assessment alpha-glucosidase inhibition, glycaemic index, oxidative stress markers, enzymatic activities, lipid profiles, and electrolyte concentrations serves distinct analytical purposes. Antioxidant capacity measurements elucidate the organism's ability to counteract oxidative stress, a pivotal factor in diabetic cardiovascular complications. Evaluating alpha-amylase and alpha-glucosidase inhibition provides insights into glycemic control mechanisms crucial for managing hyperglycemia. Assessments of oxidative stress markers provides insights into cellular damage pathways and antioxidant defense mechanisms. Oxidative stress and inflammation causes damage to cardio myocytes and release of enzymes like CKMB, LDH and natriuretic peptides ANP and BNP which are indicators of heart failure, assessing these markers provide insights to heart functionalizing and response of treatment. Additionally, lipid profile analysis and electrolyte measurements contribute to understanding metabolic homeostasis and cardiovascular health. Integration of these findings facilitates a comprehensive understanding of the multifaceted pathophysiology of diabetic cardiomyopathy, guiding targeted therapeutic strategies to mitigate cardiac dysfunction in diabetic patients. These details have been added in discussion.
Comment 6: Relevant information is missing in the methodology sections, e.g. in 2.5 for the UV/VIS dectector the wavelength used in the assays is missing, in 2.6 the operating conditions of the mass spectrometric detector are missing. Likewise, the criteria for comparison of the mass spectra obtained with the mass spectra in the Wiley database. It does not indicate the injection volume used and the injection mode whether it was Split or Splitless. As discussed above you should open each section of the methodology
Response 6: The wavelength used for the UV/VIS is already mention in the manuscript (Page 15, Line 502). The method for HPLC and GCMS have been added in detail with injection volume and mode (Page 15, Line 486 to 512).
Comment 7: The discussion should follow a logical structure including: A) Summary of the main findings. B) Comparison with previous studies. C) Explanation of biological mechanisms. D) Clinical implications. E) Limitations of the study. F) Suggestions for future research.
Response 7: A) The summary of main findings has been incorporated in discussion (Page 10, Line 261 to 279). B) The comparison with previous studies is already included in the manuscript (Line 279, 306, 316, 331, 337, 342, 352, 377, 383, 390). C) Biological mechanisms are already written in manuscript (Line 274 to 279, 285, 311, 312, 319, 397, 370, 283, 323, 336, 355, 368, 398, 344) D) Clinical implication is already written (Line 306, 307). E) Limitations of the study and F) Suggestions for future research has been added as advised (Line 401 to 406).
Comment 8: Ensure that paragraphs are logically connected. Use transitional sentences to guide the reader from one point to the next.
Response 8: The paragraphs have been rearranged to enhance the flow and coherence of the text
Comment 9: Check that all statements are supported by appropriate citations and check that references are correctly formatted according to journal guidelines.
Response 9: All statements are supported by appropriate citations and references are correctly formatted according to journal guidelines.
Comment 10: You should connect the introduction to the results: Make sure you clearly connect the findings discussed with the objectives stated in the introduction and with the results presented. This helps to reinforce the coherence of the manuscript.
Response 10: The suggested changes have been implemented in the manuscript. The introduction has been revised to align with the results. The findings have been integrated into the discussion section in relation to the study objectives, as advised.
Comment 11: emphasis the key results of your study at the end of each subsection to remind the reader of the most important findings.
Response 11: The suggested changes have been implemented in the manuscript (Line 121, 122, 134, 135, 144, 145, 153, 154, 162, 168, 176, 179, 189, 191, 199, 201)
Comment 12: The end of the discussion should include a paragraph with the strengths and weaknesses of the work you have done and presented.
Response 12: The recommended revisions have been integrated into the manuscript (Page 13, Line 401 to 406).
Reviewer 3 Report
Comments and Suggestions for Authors
1Please check the name of the first author.
2The abstract needs to be revised: you need to add information about the relevance of the research and cite the results of your own research with data. Now it is simply a statement of the fact that what was done.
3In the introduction, it is necessary to describe the relevance of phytotherapy for these cases and justify why the choice fell on Artemisia vulgaris. What was already known about it. It is also interesting to justify the choice of roots as raw material, because the generally accepted raw material for wormwood is herb.
4How and by whom was the raw material identified? Add information about its origin. Why was the root chosen as raw material?
5The extraction method is not very clear. It needs to be clarified. How completeness of methanol evaporation was checked? If it was pure methanol, then what kind of solution is spoken about in lines 90-91?
6Line 94 shows “bioactive elements”. What does this mean here?
7Methods of phytochemical study, for example, paragraphs 2.3, 2.4, 2.7, etc. or not described at all, or very briefly. In order to understand what we are talking about, you need to additionally search for this information in the references, which is very inconvenient when reading. Therefore, it is necessary to briefly describe all the methods.
8It is necessary to provide information about the reagents that were used.
9The Figure 1 is not of particular importance, the numerical data are given in the text, so it is advisable to remove it.
1It is not clear how the data in Table 1 were obtained. There is no information in either the methods or the results.
1The Figure 2 is of poor quality. There are too many markings on it, so it is difficult to understand it. It may be advisable to remove the retention time and present only the peaks of substances with titles.
1The Figure 3 (line 209) and the Figure5 (line 214-215) are completely absent. What are these inscriptions for?
1What information does the Figure 4 give? It is logical that the higher the concentration, the greater the antioxidant property.
1All Latin plant names should be in italics.
1The phytochemical part of the manuscript is not clearly presented and hardly discussed.
1In the discussion, some parts are not related to your results, so they need to be connected with your results.
1 In the conclusions, add that the root extract was studied, not the plant in general. You should also add the phytochemical results and more specific your data and results. Now it is quite general.
Author Response
Comment 1: Please check the name of the first author.
Response 1: The name of the first author is ‘Liza’. The author doesn’t have the family name as per her official name. Therefore, we used the same first name in place of family name as well. However, it is requested to consider the author’s name as just ‘Liza’.
Comment 2: The abstract needs to be revised: you need to add information about the relevance of the research and cite the results of your own research with data. Now it is simply a statement of the fact that what was done.
Response 2: Abstract has been revised (Page 1, Line 11, 12, 13, 14) and results have been cited (Page 1, Line 21 to 31).
Comment 3: In the introduction, it is necessary to describe the relevance of phytotherapy for these cases and justify why the choice fell on Artemisia vulgaris. What was already known about it. It is also interesting to justify the choice of roots as raw material, because the generally accepted raw material for wormwood is herb.
Response 3: The relevance of phytotherapy and has been added in the introduction (Page 2, Line 63 to 65) and (Page 2, Line 69 to 70). The selection for roots has been added (Page 2, Line 73 to 76). There was a lot of literature available on the aerial parts of Artemisia vulgaris and so that was the novelty of this study that it explored for the first time the effect of Artemisia vulgaris roots against Diabetic cardiomyopathy. Previously known facts include (1) high phenolic and flavonoid content in roots than leaf, (2) greater antioxidant potential of roots than leaf, demonstrated by high DPPH radical scavenging percentage and low ic50 value in roots (The more the percentage inhibition, the lesser the ic50 value), (3) greater anti diabetic potential of roots than leaf, demonstrated by high percentage inhibition of alpha amylase and alpha glucosidase and low ic50 value in roots. These findings motivated our decision to investigate the roots of Artemisia vulgaris (Sharma et al., 2023),
Comment 4: How and by whom was the raw material identified? Add information about its origin. Why was the root chosen as raw material?
Response 4: The raw material was identified by an expert botanist, associate professor of department of botany at Government College University Faisalabad. The identification number of this specimen is 405-bot-24. This has been added in manuscript as well (Page 14, Line 445, 446). The origin of Artemisia vulgaris and the reason of roots selection has been incorporated in manuscript (Page 2, Line 71 to 76) The roots of this plant are also been utilized for medicinal purposes, prompting our selection to investigate its potential in diabetic cardiomyopathy (Siwan et al., 2022). Additionally, the roots have exhibited anti-hyperlipidemic effects (Page 2, Line 76), and given our animal model's high-fat diet induction, our primary focus was to identify treatments capable of reducing lipid levels and preserving cardiac health by mitigating lipotoxicity—a critical factor in diabetic cardiomyopathy (Khan et al., 2015).
Comment 5: The extraction method is not very clear. It needs to be clarified. How completeness of methanol evaporation was checked? If it was pure methanol, then what kind of solution is spoken about in lines 90-91?
Response 5: This is an interesting query; we appreciate your question. The missing details in extraction method has been added in the manuscript (Page 14, Line 450, 451). The solution was the concentrated filtrate, which was partially condensed. To ensure complete evaporation of methanol and achieve full condensation of the solution, it was placed in an incubator. While methanol can evaporate at room temperature due to its volatility, using the incubator ensured thorough evaporation and complete dryness of the solution. The methods of Sharma et al., 2023 were used to make extract with few modifications. This reference is cited in manuscript as well (Page 14, Line 451).
Comment 6: Line 94 shows “bioactive elements”. What does this mean here?
Response 6: Bioactive elements in that line referred to specific compounds within plants that exhibit biological activity and have potential health benefits. These elements can include: alkaloids, saponins, polyester, terpenoids, phenols and flavonoids.
Comment 7: Methods of phytochemical study, for example, paragraphs 2.3, 2.4, 2.7, etc. or not described at all, or very briefly. In order to understand what we are talking about, you need to additionally search for this information in the references, which is very inconvenient when reading. Therefore, it is necessary to briefly describe all the methods.
Response 7: We apologize for any inconvenience. The methodology asked for given sections has been added in detail (Page 14 to 15, Line 453 to 532). I hope this significant addition can help this manuscript meet the standard of publication.
Comment 8: It is necessary to provide information about the reagents that were used.
Response 8: We appreciate your suggestion, the details of the reagents has been mentioned in materials and methods (Page 13, Line 408 to 415).
Comment 9: The Figure 1 is not of particular importance, the numerical data are given in the text, so it is advisable to remove it.
Response 9: That particular figure has been removed.
Comment 10: It is not clear how the data in Table 1 were obtained. There is no information in either the methods or the results.
Response 10: The method to obtain the results described in table 1 has been added (Page no 14, Line 453 to 467)
Comment 11: The Figure 2 is of poor quality. There are too many markings on it, so it is difficult to understand it. It may be advisable to remove the retention time and present only the peaks of substances with titles.
Response 11: We appreciate your suggestion. The retention times were documented to enhance the reproducibility of our research findings, playing a pivotal role in precisely quantifying the presence of compounds within the extract. These times serve as critical reference points, ensuring consistency and reliability in our analytical approach. By accurately determining when each compound elutes from the chromatographic system, we can confidently calculate their respective concentrations. This meticulous measurement not only validates the reliability of our experimental data but also underscores the significance of retention times in facilitating robust scientific conclusions. Thus, their precise determination remains integral to the accuracy and credibility of our research outcomes.
Comment 12: The Figure 3 (line 209) and the Figure5 (line 214-215) are completely absent. What are these inscriptions for
Response 12: That might be due to software issue, all the labeled figures were present in the manuscript, however we make sure again that no figure should be missing in the manuscript.
Comment 13: What information does the Figure 4 give? It is logical that the higher the concentration, the greater the antioxidant property
Response 13: Yes, it is logical that the higher the concentration the greater the antioxidant (Mustafa et al., 2021). This reference has been added in the manuscript as well. The DPPH radical assay measures antioxidant potential by assessing how well substances can neutralize DPPH radicals. Higher concentrations of extracts mean more antioxidants are available to react with DPPH radicals, leading to greater reduction in color change or absorbance, indicating stronger antioxidant activity. Thus, higher extract concentrations generally correlate with higher antioxidant potential in this assay.
Comment 14: All Latin plant names should be in italics.
Response 14: All Latin plant names are written in italics.
Comment 15: The phytochemical part of the manuscript is not clearly presented and hardly discussed
Response 15: The entire discussion revolves around the phytochemical components and their possible mechanisms for treating diabetic cardiomyopathy. Specifically, we focused on the compounds identified through HPLC and GCMS (Page 11, Line 280 to 285, 289 to 291, 295, 296, 301, 313, 318, 321), (page 12, Line 334, 335, 344, 348, 354, 357, 363, 368 to 371, 380) and (Page 13, Line 390).
Comment 16: In the discussion, some parts are not related to your results, so they need to be connected with your results.
Response 16: All parts have been connected with the results.
Comment 17: In the conclusions, add that the root extract was studied, not the plant in general. You should also add the phytochemical results and more specific your data and results. Now it is quite general.
Response 17: Thank you for highlighting this point. Conclusion has been changed as advised (Page 17, Line 597 to 602).
References
Sharma, Khaga Raj, and Saroj Adhikari. "Phytochemical analysis and biological activities of Artemisia vulgaris grown in different altitudes of Nepal." International Journal of Food Properties 26, no. 1 (2023): 414-427.
Siwan, Deepali, Dipali Nandave, and Mukesh Nandave. "Artemisia vulgaris Linn: An updated review on its multiple biological activities." Future Journal of Pharmaceutical Sciences 8, no. 1 (2022): 47.
Khan, K. Abedulla. "A preclinical antihyperlipidemic evaluation of Artemisia vulgaris root in diet induced hyperlipidemic animal model." Int. J. Pharmacol. Res 5 (2015): 110-114.
Mustafa, Imtiaz, Muhammad Naeem Faisal, Ghulam Hussain, Humaira Muzaffar, Muhammad Imran, Muhammad Umar Ijaz, Muhammad Umar Sohail, Arslan Iftikhar, Arslan Shaukat, and Haseeb Anwar. "Efficacy of Euphorbia helioscopia in context to a possible connection between antioxidant and antidiabetic activities: a comparative study of different extracts." BMC Complementary Medicine and Therapies 21 (2021): 1-12.
Round 2
Reviewer 3 Report
Comments and Suggestions for Authors
In my opinion the issue with the figures wasn't solved, but it's so critical.
The manuscript was really improved.